# Chain-of-Thought Reasoning in Tabular Language Models

**Mingyu Zheng**[1,2†], **Yang Hao**[3], **Wenbin Jiang**[3] , **Zheng Lin**[1,2‡],
**Yajuan Lyu**[3], **Qiaoqiao She**[3], **Weiping Wang**[1]

[1]Institute of Information Engineering, Chinese Academy of Sciences, Beijing, China
[2]School of Cyber Security, University of Chinese Academy of Sciences, Beijing, China
[3]Baidu Inc, Beijing, China
{zhengmingyu,linzheng,wangweiping}@iie.ac.cn
{haoyang03,jiangwenbin,lvyajuan,sheqiaoqiao}@baidu.com

## Abstract

Tabular mathematical reasoning task requires models to perform multi-step operations including information look-up and numerical calculations, based on heterogeneous data from tables and questions. Existing solutions tend to extend chain-of-thought (CoT) reasoning into powerful large language models (LLMs) to promote multi-hop mathematical reasoning. However, it can be extremely difficult to apply such LLM-based approaches under scenarios of privatization deployment or limited resources. To address this problem, we revisit small-scale tabular language models (TaLMs) and extend chain-of-thought reasoning into TaLMs for the first time. Specifically, we propose a novel framework, **TaCo**, which coordinates two TaLMs responsible for CoT generation and answer inference, respectively. Besides, our framework can be combined with an external calculator to enhance accurate numerical calculations. On the TABMWP dataset, TaCo outperforms the state-of-the-art ChatGPT by 9.55% (82.60%→92.15% in accuracy) with much less parameters (0.8B).[1]

## 1 Introduction

Tabular mathematical reasoning task aims at answering math questions based on heterogeneous tabular and textual data, which can provide users with insights from tables containing valuable figures (Lu et al., 2023b; Zhu et al., 2021; Chen et al., 2021b). This task highlights the demand for multi-step mathematical reasoning including information look-up and numerical calculations. For example, given the table and the question in Figure 1, we firstly need to count how many numbers are in the table, then add all the numbers together to get the sum of baskets, and finally compute the mean of the sum.

| Day | Number of baskets |
|---|---|
| Thursday | 49 |
| Friday | 48 |
| Saturday | 51 |
| Sunday | 54 |
| Monday | 37 |
| Tuesday | 49 |

**Question:** Reagan jotted down how many baskets she made during basketball practice each day. What is the mean of the numbers?
**Solution:**
Read the numbers from the table.
49, 48, 51, 54, 37, 49
First, count how many numbers are in the group.
There are 6 numbers.
Now add all the numbers together:
49 + 48 + 51 + 54 + 37 + 49 = 288
Now divide the sum by the number of numbers:
288 ÷ 6 = 48
The mean is **48**.
**Answer: 48**

Figure 1: An example from the TABMWP dataset. To solve the problem, the model needs to perform multi-step mathematical reasoning based on the table and the question.

Considering the inherent demand for multi-step operations, existing studies tend to extend chain-of-thought (CoT) reasoning (Wei et al., 2022; Wang et al., 2023a; Kojima et al., 2022; Zhang et al., 2022) into powerful Large Language Models (LLMs) (Brown et al., 2020; Chowdhery et al., 2022; Thoppilan et al., 2022; Chen et al., 2021a) to promote multi-hop mathematical reasoning. As depicted in Figure 2 (b), this paradigm prompts LLMs with several in-context examples containing CoT demonstrations to elicit intermediate reasoning steps before inferring the final answer.

Though the combo of LLM and CoT has achieved great performance, such LLM-based methods may not be a feasible approach in some real-world scenarios. For instance, it is financially expensive to satisfy the high computational requirements, the storage capacity and the desired bandwidth of LLMs, which makes it a challenge for

---

[1]The code will be released at https://github.com/SpursGoZmy/TaCo

[†]This work was done during an internship at Baidu Inc.

[‡] Corresponding author: Zheng Lin.

individual users or small organizations to utilize LLMs in their applications (Strubell et al., 2019; Bender et al., 2021). In consideration of the data security, enterprises may also seek privatization deployments where private data is not allowed to be processed by third-party LLM APIs. What's more, despite the fact that many pre-trained tabular language models have been developed (Liu et al., 2022; Herzig et al., 2020; Wang et al., 2021; Dong et al., 2022), their CoT reasoning ability has not been thoroughly investigated and it could be inadequate for solving the tabular mathematical reasoning task. As a result, an alternative approach, with lower costs and competitive CoT reasoning ability, is needed.

To accomplish this goal, we revisit small-scale tabular language models (TaLMs) and initiatively explore the chain-of-thought reasoning in TaLMs. Specifically, we propose a novel framework named **TaCo**, which coordinates two **TaLMs** that are responsible for **CoT** generation and answer inference, respectively. Given the input table and question, the first TaLM is fine-tuned to generate intermediate reasoning steps. Based on the original input and generated reasoning steps, the second TaLM is fine-tuned to infer the final answer. To alleviate the weakness of TaLMs in solving mathematical expressions, TaCo is also combined with an external calculator which is used to perform math calculations and fix incorrect results in the output reasoning steps.

To verify the effectiveness of the proposed method, we conduct comprehensive experiments on the TABMWP (Lu et al., 2023b) dataset, which is the latest math word problem benchmark over tabular data and provides detailed chain-of-thoughts to solve the problem step by step. Experimental results reveal that TaCo explores a new and promising paradigm for tabular mathematical reasoning, which is illustrated in Figure 2 (c). Compared with traditional fine-tuned TaLMs, TaCo improves the accuracy of recent TAPEX model by 29.76%. Compared with LLM-based approaches, TaCo outperforms the state-of-the-art ChatGPT by 9.55% (82.60%→92.15%) with much less parameters (0.8B). Moreover, we conduct ablation studies to analyse contributions of different parts in the framework. The detailed error analysis is also performed to provide insights for future improvements.

To summarize, we conclude our contributions as follows:

- To the best of our knowledge, we explore the chain-of-thought reasoning in TaLMs for the first time, and advocate a new and promising paradigm for tabular mathematical reasoning, especially under scenarios where LLM-based methods are not feasible.

- We propose a novel framework, TaCo, which coordinates two TaLMs responsible for CoT generation and answer inference, respectively. It is also integrated with a calculator to enhance accurate numerical calculations.

- Our method can boost the performance of small-scale TaLMs and surpasses the state-of-the-art ChatGPT by 9.55% on TABMWP benchmark with much less parameters (0.8B).

## 2 Pilot Experiment

Before diving into the specific method, we present a pilot experiment on the TABMWP dataset to answer two important questions: (i) Do existing pre-trained generative TaLMs possess chain-of-thought reasoning ability? (ii) Whether generative TaLMs can benefit from chain-of-thoughts when predicting the final answer. We select the state-of-the-art TAPEX model (Liu et al., 2022) for experiments, which is based on the encoder-decoder language model BART (Lewis et al., 2020) and is additionally pre-trained on the tabular data. We consider two model sizes: TAPEX-base (140M) and TAPEX-large (400M).

Experiments are conducted in three different settings, i.e., vanilla, zero-shot CoT and gold CoT. For the "vanilla" setting, the pre-trained TAPEX model $f(\cdot)$ autoregressively generates the answer $a$ based on the table $t$ and the question $q$, i.e., $a = f(t, q)$. For the "zero-shot CoT" setting, we follow Kojima et al. (2022) to evaluate the CoT reasoning of the TAPEX. Specifically, a trigger sentence $p_1$ is appended to the question in order to ask the TAPEX to output intermediate reasoning steps $s$, i.e., $s = f(t, q, p_1)$. Then, given the original input and the generated CoT, another trigger sentence $p_2$ is appended to make the TAPEX output the final answer $a$, i.e., $a = f(t, q, p_1, s, p_2)$. For $p_1$, we try various templates such as "Let's think step by step" and report best results. For $p_2$, we intuitively select "As a result, the answer is" as the trigger sentence. For the "gold CoT" setting, we replace

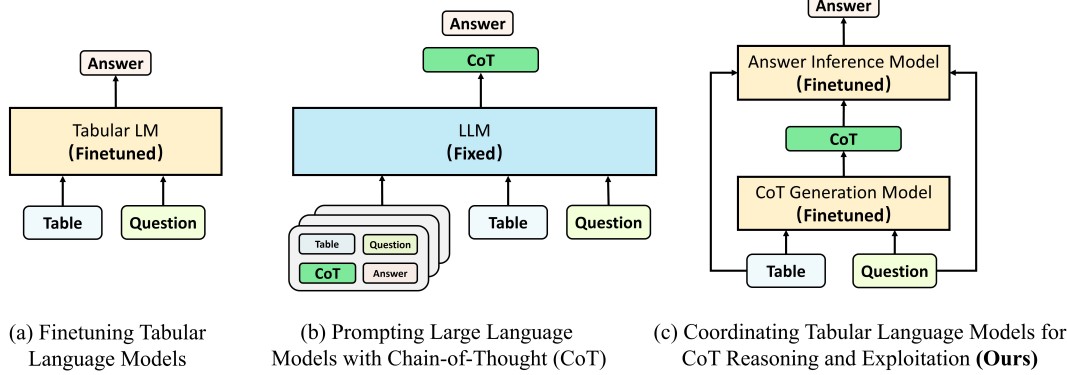

(a) Finetuning Tabular Language Models

(b) Prompting Large Language Models with Chain-of-Thought (CoT)

(c) Coordinating Tabular Language Models for CoT Reasoning and Exploitation **(Ours)**

Figure 2: Different paradigms for tabular mathematical reasoning.

generated reasoning steps with annotated ones and other procedures are same as "zero-shot CoT".

| Pre-trained TaLMs | Acc-Dev | Acc-Test |
|---|---|---|
| TAPEX-base (vanilla) | 15.66 | 15.69 |
| TAPEX-large (vanilla) | 18.41 | 18.59 |
| TAPEX-base (zero-shot CoT) | 15.30 | 15.25 |
| TAPEX-large (zero-shot CoT) | 18.25 | 17.94 |
| TAPEX-base (gold CoT) | 40.54 | 39.99 |
| TAPEX-large (gold CoT) | 47.48 | 48.01 |

Table 1: Pilot experimental results of pre-trained TAPEX under different settings. "Acc-Dev" and "Acc-Test" represents accuracy on the development set and the test set respectively.

From the results in Table 1, we can see that the TAPEX with "zero-shot CoT" setting performs even worse than the vanilla one, which shows that the small-scale TAPEX is not a decent zero-shot reasoner like LLMs and does not possess CoT reasoning ability. This is also consistent with findings from previous CoT studies (Wei et al., 2022; Ho et al., 2023). After inspecting the model outputs, we find that the pre-trained TAPEX model cannot follow the instruction to generate reasoning steps. In most cases, it directly generates the answer or illogical texts. However, given the annotated "gold CoT", the model achieves a remarkable performance gain. For instance, the accuracy of TAPEX-large on test set increases from 18.59% to 48.01%. This demonstrates that CoT reasoning steps are beneficial to TAPEX when inferring the correct answer and it encourages us to further elicit CoT reasoning ability of TaLMs by finetuning.

## 3 Method

Based on observations in Section 2, we propose the TaCo framework for tabular mathematical reasoning. It includes two training stages: (i) CoT

generation and (ii) answer inference, where two generative TaLMs with the same architecture are fine-tuned independently with different inputs and outputs. In this section, we introduce the framework with the TAPEX model as selected backbones, but it should be noted that TaCo is compatible with arbitrary generative TaLMs to boost their performance. The overview of TaCo framework is illustrated in Figure 3.

### 3.1 CoT Generation

In the CoT generation stage, a TAPEX model is fine-tuned to generate a solution which consists of multiple reasoning steps to solve the problem. Given an input table $T$ with $M$ rows $\{R_i\}_{i=1}^M$ and $N$ column headers $\{c_j\}_{j=1}^N$, the TAPEX will linearize the table into a flattened text sequence $T^* = $ [HEAD] $:$ $c_1 \mid \cdots \mid c_N$ [ROW] $1$ $:$ $R_1 \mid$ [ROW] $2$ $:$ $R_2 \mid \cdots \mid R_M$, where [HEAD] and [ROW] are special tokens used to indicate the region of column headers and rows, respectively. The number after [ROW] represents different row index and the vertical bar "|" separates headers or cells in different columns. For instance, the table in Figure 1 will be linearized into the following sequence:

> col : Day | Number of baskets row 1 : Thursday | 49 row 2 : Friday | 48 ... row 6 : Tuesday | 49

The resulting sequence $T^*$ will be concatenated with the textual context, which includes a question $Q$ and a trigger sentence $P$. Based on the concatenated input, the probability of generating the target solution $S$ is computed as follows:

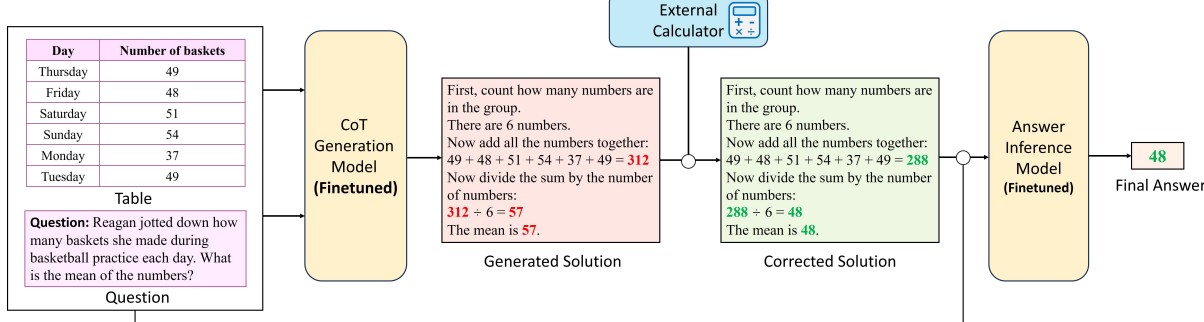

Figure 3: Overview of the TaCo framework, with the table and the question in Figure 1 as a running example.

$$p(S|T^*, Q, P) = \prod_{i=1}^{L} p_\theta(S_i|T^*, Q, P, S_{<i}) \quad (1)$$

where $L$ is the length of target solution. We select "Let's think step by step" as the trigger sentence $P$ since it gives the best performance in pilot experiments.

After generating a potential solution $\bar{S}$, we find that $\bar{S}$ often contains some numerical calculation errors. This is often the case with language models because TaLMs and even LLMs are not suitable for actually solving mathematical expressions (Chen et al., 2022). Take the generated solution in Figure 3 as an example. Though the model generates plausible reasoning steps, calculation results among these steps are all wrong (in red color), e.g., "49 + 48 + 51 + 54 + 37 + 49 = 312". Such calculation errors will accumulate to the last reasoning step and seriously mislead the answer inference model into predicting the false answer.

To mitigate the influence of calculation mistakes, we introduce an arithmetic calculator $g(\cdot)$ to solve mathematical expressions of "+,-,×,÷" in the generated solution $\bar{S}$ and output the corrected solution $\hat{S} = g(\bar{S})$. Concretely, we extract equation strings in $\bar{S}$ using regular expressions and calculate their results using the Python `eval` function. Since multiple equations may exist in one solution and one equation could also refer to results of previous equations, the calculation result of each equation is propagated to the following equations by string replacing. As we can see from Figure 3, original wrong results in $\bar{S}$ are successfully fixed and are replaced with correct results (in green color), e.g., "49 + 48 + 51 + 54 + 37 + 49 = 288".

## 3.2 Answer Inference

In answer inference stage, another TAPEX model is fine-tuned to generate the final ansewr based on the original input and the annotated solution $S$. Similar with the CoT generation stage, the probability of generating target answer $A$ is computed by:

$$p(A|T^*, Q, P, S) = \prod_{i=1}^{N} p_\theta(A_i|T^*, Q, P, S, A_{<i})$$
$$(2)$$

where $N$ is the length of target answer. During the inference phase, the annotated solution is replaced with the corrected solution $\hat{S}$ to output the predicted answer $\bar{A}$. Both CoT generation model and answer inference model are trained with a standard language modeling objective.

## 4 Experiments

### 4.1 Dataset and Evaluation Metric

Experiments are conducted on the TABMWP (Lu et al., 2023b) dataset, a recent large-scale benchmark which is constructed from grade-level math curricula and contains 38,481 math word problems with the tabular context. Beside the gold answers, TABMWP also provides detailed step-by-step solutions to solve the problems, which can be utilized as chain-of-thoughts to finetuning TaLMs. There are two question-types in the TABMWP: 28,719 *free-text* questions with integer answers (INT) and decimal answers (DEC), and 9,712 *multi-choice* questions with extractive text answers (EXTR), boolean text answers (BOOL) and other text answers (OTH). Statistics of each split are shown in the Table 2. The test set contains 7,686 questions in total. Among them, 74.08% are INT (4,529) and DEC (1165) questions, and 25.92% are DEC (1,165), EXTR (987) and OTH (105) questions. Thus, INT and

|  | Train | Dev | Test | Total |
|---|---|---|---|---|
| # of questions | 23,059 | 7,686 | 7,686 | 38,431 |
| # of free-text | 17,135 | 5,710 | 5,694 | 28,719 |
| # of multi-choice | 5,744 | 1,976 | 1,992 | 9,712 |
| # of tables | 22,620 | 7,546 | 7,549 | 37,644 |
| # of solutions | 21,623 | 7,365 | 7,378 | 35,442 |

Table 2: Dataset statistics of TABMWP.

DEC questions are more essential for the overall accuracy. Given the predicted answer and the ground truth, we employ the exact match accuracy as the metric and use the official evaluation script to evaluate the model performance.

## 4.2 Implementation Details

**Implementations.** Our framework is implemented with Pytorch (Paszke et al., 2019). We mainly employ the TAPEX (Liu et al., 2022) as the backbone TaLM in the proposed framework. We also replace TAPEX with UnifiedQA (Khashabi et al., 2020) for the ablation study. Various model sizes are included to present more valid evaluation across different model capacities. Both CoT generation model and answer inference model are optimized by AdamW (Loshchilov and Hutter, 2019). We use validation set for the model selection and manually tune hyper-parameters, and evaluate the best model on the test set. For CoT generation, we adopt the beam search decoding with the beam size of 3. For answer inference, we adopt the greedy decoding. Hyper-parameter configurations for best-performing models and more implementation details are shown in the Table 6 and Table 7.

**Baselines.** (1) *Pre-trained and Fine-tuned language models:* We develop **TAPEX** (Liu et al., 2022) and **UnifiedQA** (Khashabi et al., 2020) in both pre-trained and fine-tuned settings to predict the final answer. TAPEX is the state-of-the-art BART-based (Lewis et al., 2020) TaLM which is pre-trained on the tabular data to mimic a SQL executor. UnifiedQA is a T5-based (Raffel et al., 2020) QA model which is pre-trained on 8 QA datasets of multiple formats. We consider three model sizes for UnifiedQA: small (60M), base (220M) and large (770M). Given the flattened table and question, both TAPEX and UnifiedQA can generate the answer text autoregressively. (2) *Large language models:* We consider **GPT-3** (Brown et al., 2020), **Codex** (Chen et al., 2021a) and **Chat-GPT** with the standard few-shot and zero-shot prompting. ChatGPT is based on the *gpt-3.5-turbo* engine. Numbers of in-context examples and se-

lection strategies for few-shot prompting are listed in Table 8. (3) *Large language models with CoT prompting:* Beside standard prompting, we also consider above LLMs with the chain-of-thought prompting. **PromptPG** (Lu et al., 2023b) utilizes the policy gradient method to select in-context examples for test samples when constructing the prompt for LLMs. **PoT** (Chen et al., 2022) proposes the "program-of-thoughts", which exploits Codex to generate the text and Python program for math computations. The generated program is executed by a program interpreter to output the final answer. The "Heuristic guess" is a baseline from the TABMWP paper. For multi-choice questions, it randomly selects one from the given options with even probabilities. For free-text questions, it randomly chooses one number from the question or the table as the prediction.

## 4.3 Main Results

Table 3 demonstrates main experimental results on the TABMWP dataset. For TAPEX, UnifiedQA and ChatGPT baselines, we report results based on our implementation. For other baselines, we report published results from original papers (Lu et al., 2023b; Chen et al., 2022).

From the results in Table 3, we can find that: (1) With two TAPEX-large models as backbones, the TaCo framework establishes a new state-of-the-art accuracy of 92.15% on the TABMWP test set, outperforming the previous best model ChatGPT with CoT prompting by 9.55%, which demonstrates the effectiveness of the proposed method. Notably, compared with LLMs such as GPT-3 and Codex, the parameters in TaCo framework are much less (0.8B), which brings lower costs for application deployments. (2) Compared with LLM-based approaches with the standard few-shot prompting, fine-tuned TAPEX and UnifiedQA can achieve competitive results. For instance, the fine-tuned TAPEX-large even performs better than GPT-3 and Codex. However, when combined with the CoT prompting, LLM-based methods are significantly better than fine-tuned small-scale language models, which shows that the CoT prompting plays an important role in the tabular mathematical reasoning task. By contrast, the TaCo framework extends the CoT reasoning into TaLMs for the first time, and improves the performance of TAPEX-base and TAPEX-large model by 29.19% and 29.76%, respectively.

| Model | Acc-Dev | Acc-Test | Question Types | | Answer Types | | | | | Grades | |
|---|---|---|---|---|---|---|---|---|---|---|---|
| | | | FREE | MC | INT | DEC | EXTR | BOOL | OTH | 1-6 | 7-8 |
| *Heuristic baselines* | | | | | | | | | | | |
| Heuristic guess | - | 15.29 | 6.71 | 39.81 | 8.37 | 0.26 | 30.80 | 51.22 | 26.67 | 17.55 | 12.27 |
| Human performance | - | 90.22 | 84.61 | 93.32 | 84.95 | 83.29 | **97.18** | 88.69 | **96.20** | **94.27** | 81.28 |
| *Pre-trained LM* | | | | | | | | | | | |
| TAPEX-base | 15.66 | 15.69 | 7.29 | 39.71 | 8.63 | 2.06 | 34.95 | 47.11 | 20.95 | 18.6 | 11.81 |
| TAPEX-large | 18.41 | 18.59 | 8.80 | 46.59 | 10.62 | 1.72 | 46.91 | 48.11 | 30.48 | 22.65 | 13.18 |
| UnifiedQA-small | 10.71 | 12.18 | 1.18 | 43.62 | 1.37 | 0.43 | 38.7 | 49.78 | 37.14 | 15.57 | 7.65 |
| UnifiedQA-base | 12.10 | 14.56 | 4.60 | 43.02 | 5.28 | 1.97 | 37.08 | 50.11 | 38.1 | 17.14 | 11.11 |
| UnifiedQA-large | 14.00 | 14.06 | 3.37 | 44.63 | 4.02 | 0.86 | 40.53 | 50.22 | 35.24 | 17.21 | 9.87 |
| *Fine-tuned LM* | | | | | | | | | | | |
| TAPEX-base | 57.10 | 56.39 | 48.33 | 79.42 | 56.33 | 17.25 | 90.37 | 67.78 | 76.19 | 65.17 | 44.67 |
| TAPEX-large | 62.28 | 62.39 | 55.50 | 82.08 | 64.21 | 21.63 | 96.47 | 65.78 | 77.14 | 71.32 | 50.47 |
| UnifiedQA-small | 35.79 | 34.82 | 27.99 | 54.32 | 33.94 | 4.89 | 52.99 | 53.89 | 70.48 | 42.23 | 24.93 |
| UnifiedQA-base | 51.89 | 51.08 | 42.10 | 76.76 | 49.83 | 12.02 | 89.16 | 63.33 | 75.24 | 59.03 | 40.48 |
| UnifiedQA-large | 59.35 | 59.26 | 51.62 | 81.12 | 60.68 | 16.39 | 92.20 | 69.44 | 77.14 | 67.11 | 48.80 |
| *LLM* | | | | | | | | | | | |
| GPT-3 (zero-shot) | - | 56.96 | 53.57 | 66.67 | 55.55 | 45.84 | 78.22 | 55.44 | 54.29 | 63.37 | 48.41 |
| GPT-3 | - | 57.13 | 54.69 | 64.11 | 58.36 | 40.40 | 75.95 | 52.41 | 53.02 | 63.10 | 49.16 |
| Codex | - | 59.40 | - | - | - | - | - | - | - | - | - |
| ChatGPT | 64.12 | 65.52 | 65.84 | 64.61 | 66.55 | 63.09 | 74.67 | 54.67 | 55.24 | 69.75 | 59.88 |
| *LLM+CoT* | | | | | | | | | | | |
| GPT-3 (zero-shot) | - | 57.61 | 54.36 | 66.92 | 55.82 | 48.67 | 78.82 | 55.67 | 51.43 | 63.62 | 49.59 |
| GPT-3 | - | 62.92 | 60.76 | 69.09 | 60.04 | 63.58 | 76.49 | 61.19 | 67.30 | 68.62 | 55.31 |
| Codex | - | 65.20 | - | - | - | - | - | - | - | - | - |
| PromptPG | - | 68.23 | 66.17 | 74.11 | 64.12 | 74.16 | 76.19 | 72.81 | 65.71 | 71.20 | 64.27 |
| Codex-SC | - | 75.40 | - | - | - | - | - | - | - | - | - |
| PoT | - | 73.20 | - | - | - | - | - | - | - | - | - |
| PoT-SC | - | 81.80 | - | - | - | - | - | - | - | - | - |
| ChatGPT | 82.49 | 82.60 | 80.89 | 87.50 | 79.36 | 86.87 | 81.86 | **94.00** | 84.76 | 82.68 | 82.51 |
| *Ours* | | | | | | | | | | | |
| TaCo (TAPEX-base) | 86.12±0.13 | 85.58±0.14 | 85.53 | 85.74 | 85.29 | 86.44 | 93.31 | 77.89 | 81.90 | 87.43 | 83.12 |
| **TaCo (TAPEX-large)** | **92.91±0.17** | **92.15±0.13** | **91.69** | **93.47** | **92.54** | **88.41** | 96.05 | 91.44 | 86.67 | 92.37 | **91.86** |

Table 3: Accuracy (%) on the development set and test set of TABMWP. We also report detailed accuracy on different types of questions in test set. FREE: *free-text* questions; MC: *multi-choice* questions. INT: integer answers; DEC: decimal answers; EXTR: extractive text answers; BOOL: Boolean text answers; OTH: other text answers. The best results are marked in **bold**. ± stands for standard deviation over 3 repeated experiments. If not otherwise specified, LLM baselines are in few-shot setting. "-SC" represents using self-consistency decoding strategy (Wang et al., 2023a).

(3) Among different baselines, the model performance on *free-text* questions is obviously worse than that on *multi-choice* questions, with an average difference of 21%. The reason is that, compared with *multi-choice* questions, *free-text* questions usually require more complicated numerical calculations and also do not directly provide answer options in the input. The detailed evidence is presented in the Appendix B. Nevertheless, from *pre-trained LM* to *LLM+CoT* and to the proposed TaCo framework, the performance gap between two question types gradually decreases. For instance, the accuracy gap of TaCo (TAPEX-large) framework (1.78%) is much lower than that of fine-tuned TAPEX-large (26.58%). This shows our method can obtain better generalization on two types of questions. (4) Considering questions of various answer types, the TaCo framework beats other baselines on questions with integer (INT) and decimal (DEC) answers, which may resulted from the utilization of the external calculator. ChatGPT with the CoT prompting outperforms other methods including the human baseline on questions with Boolean text answer, which may contribute to its great general semantic understanding ability. For example, judging yes/no questions based on previously generated reasoning steps. (5) Not surprisingly, all the models perform worse on questions from the grade 7-8 than that from the grade 1-6 due to the increasing difficulty. Among them, the proposed framework achieves the best accuracy than other baselines on harder questions from grade 7-8.

### 4.4 Ablation Study

We conduct ablation experiments to systematically investigate the effect of the external calculator, the progressive two-stage paradigm and the TaLM

| Settings | Dev | Test | Average Drop↓ | Question Types FREE | MC |
|---|---|---|---|---|---|
| **ours** | | | | | |
| TaCo (base) | 86.12 | 85.58 | - | 85.53 | 85.74 |
| TaCo (large) | 92.91 | 92.15 | - | 91.69 | 93.47 |
| **w/o calculator** | | | | | |
| $QT \rightarrow S \rightarrow A$ (base) | 65.21 | 64.35 | 21.07 | 56.23 | 84.55 |
| $QT \rightarrow S \rightarrow A$ (large) | 75.60 | 74.58 | 17.44 | 67.77 | 93.03 |
| **w/o two-stage paradigm** | | | | | |
| $QT \rightarrow SA$ (base) | 78.22 | 77.66 | 7.91 | 77.15 | 79.12 |
| $QT \rightarrow SA$ (large) | 84.73 | 84.25 | 8.04 | 83.95 | 85.14 |
| $QT \rightarrow AS$ (base) | 75.18 | 74.34 | 11.09 | 71.88 | 81.38 |
| $QT \rightarrow AS$ (large) | 81.45 | 81.41 | 11.10 | 80.21 | 84.84 |
| **w/o two-stage paradigm and calculator** | | | | | |
| $QT \rightarrow SA$ (base) | 59.69 | 59.41 | 26.30 | 50.86 | 83.84 |
| $QT \rightarrow SA$ (large) | 69.57 | 68.85 | 23.32 | 63.79 | 83.33 |
| $QT \rightarrow AS$ (base) | 56.43 | 54.85 | 30.21 | 45.64 | 81.17 |
| $QT \rightarrow AS$ (large) | 63.80 | 63.41 | 28.93 | 56.06 | 84.44 |
| **w/o two-stage paradigm, calculator and solution** | | | | | |
| $QT \rightarrow A$ (base) | 57.10 | 56.39 | 29.11 | 48.33 | 79.42 |
| $QT \rightarrow A$ (large) | 62.28 | 62.39 | 30.20 | 55.50 | 82.08 |

Table 4: Ablation study of the external calculator and proposed two-stage paradigm. "base" and "large" stands for model sizes of TAPEX backbone.

| Model | Dev | Test | Question Types FREE | MC |
|---|---|---|---|---|
| **w/ TAPEX** | | | | |
| TAPEX-base | 86.12 | 85.58 | 85.53 | 85.74 |
| TAPEX-large | 92.91 | 92.15 | 91.69 | 93.47 |
| **w/ UnifiedQA** | | | | |
| UnifiedQA-small | 48.32 | 48.17 | 46.45 | 53.06 |
| UnifiedQA-base | 66.32 | 65.46 | 60.70 | 79.07 |
| UnifiedQA-large | 77.44 | 76.96 | 73.50 | 86.85 |
| **fine-tuned** | | | | |
| UnifiedQA-small | 35.79 | 34.82 | 27.99 | 54.32 |
| UnifiedQA-base | 51.89 | 51.08 | 42.10 | 76.76 |
| UnifiedQA-large | 59.35 | 59.26 | 51.62 | 81.12 |

Table 5: Experiment results of TaCo framework with TAPEX and UnifiedQA as backbone, respectively.

backbone in the TaCo framework. $QT \rightarrow S \rightarrow A$ represents the proposed two-stage paradigm, which firstly generates the solution $S$ and then arrives at the final answer $A$ based on the input question $Q$, table $T$ and generated solution $S$. $QT \rightarrow SA$ and $QT \rightarrow AS$ represents one-stage paradigms, which generate the solution and the answer in different orders, respectively. $QT \rightarrow A$ stands for the vanilla fine-tuning paradigm that directly predicts the answer.

**Effect of External Calculator.** As shown in Table 4, there is a drastic performance drop for the TaCo framework (e.g., $92.15\% \rightarrow 74.58\%$) when removing the external calculator. With further observations, we find that the performance decline mainly comes from *free-text* questions which demand more numerical calculations. For instance, the accuracy of TaCo (TAPEX-large) plummets from 91.69% to 67.77%. It demonstrates the great significance of using the external calculator to reduce calculation errors in the generated solutions. Otherwise, the answer inference model is likely to be misled by the incorrect solution and arrives at the wrong answer.

**Effect of Two-stage Paradigm.** When we change the two-stage paradigm to one-stage ones, the model performance drops about 9.5%, which reveals the contribution of two-stage paradigm. We think it is challenging for single small-scale TaLM to generate correct reasoning steps and the final answer simultaneously. As a result, we delegate the CoT generation and the answer inference to two TaLMs, respectively. More importantly, one-stage paradigms cannot fully utilize the corrected CoT to change the original (wrong) answer. By contrast, the two-stage paradigm brings a second chance to re-contemplate the improved reasoning steps before making the final judgement. The similar two-stage paradigm has also been explored in recent works (Press et al., 2023; Zhao et al., 2023), where they utilize one LLM to generate the CoT to be improved, and then ask the same LLM to infer the final answer based on the improved CoT.

Comparing two one-stage paradigms, we notice that $QT \rightarrow SA$ performs better than $QT \rightarrow AS$. This shows that it may be more suitable for TaLMs to infer the final answer according to produced reasoning steps, rather than give explanations based on the predicted final answer. If we remove both the two-stage paradigm and the external calculator, the model performance would suffer a more steep decline. But it is still better than that of traditional fine-tuned models in $QT \rightarrow A$ paradigm, which validates the value of intermediate reasoning steps for TaLMs.

**Effect of TaLM Backbone.** To investigate the performance of TaCo with different backbones, we replace TAPEX with UnifiedQA as the backbone model. Related experimental results are presented in Table 5. When the backbone changes from TAPEX to UnifiedQA, the TaCo framework suffers a sharp performance drop on both *free-text* and *multi-choice* questions. For instance, even with more parameters (1.54B), the accuracy of TaCo with UnifiedQA-large on the test set (76.96%) is much lower than that with TAPEX-large (92.15%), which indicates the advantages of pre-trained tabular language models. Unlike UnifiedQA which is solely pre-trained on the unstructured textual data,

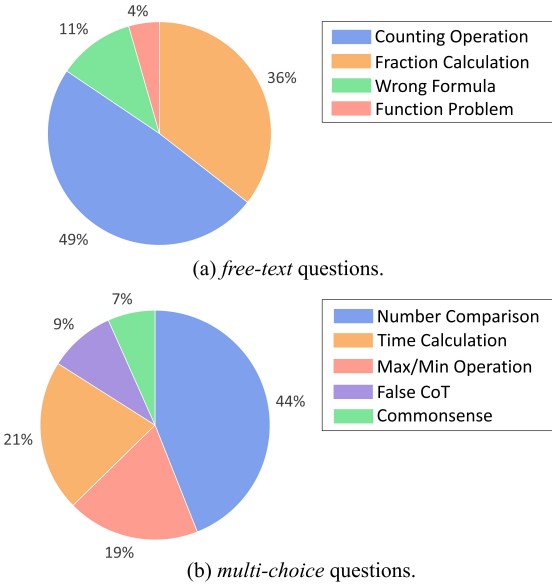

(a) *free-text* questions.

(b) *multi-choice* questions.

Figure 4: Error distributions of different question types.

TAPEX is additionally pre-trained on the tabular data and thus has a better understanding of table structures. As more powerful generative TaLMs emerge, they can be integrated into the TaCo framework to improve their performance on the tabular mathematical reasoning task.

### 4.5 Error Analysis and Case Study

As illustrated in Figure 6, for this problem that involves two multiplication and one addition operations, the TaCo framework successfully generates correct intermediate reasoning chains and finally predicts the right answer.

There are 473 *free-text* questions (78%) and 130 *multi-choice* questions (22%) for which the TaCo (TAPEX-large) gives wrong predictions. We randomly selected 100 questions of each type for error analyses. Figure 4 depicts error distributions by question types. More error instances are presented and discussed in Appendix C.

For *free-text* questions, error cases fall into the following four categories. (1) Counting operation (49%): the question requires the model to count numbers as the final answer, which is challenging for generative language models. (2) Fraction calculation (36%): the model fails to conduct fraction-related calculations such as reducing a fraction, which may be alleviated with an advanced calculator. (3) Wrong formula (11%): the CoT generation model outputs wrong formulas in the reasoning steps. (4) Function-related problem (4%): the model fails to solve problems related to the func-

tion, e.g., compute the slope of the function based on the table data.

For *multi-choice* questions, error cases can be divided into the following five types. (1) Number comparison (44%): the model cannot determine which number is larger or smaller. (2) Time calculation (21%): the model needs to perform time calculation such as compute the elapsed time between 9:15 A.M. and 11:20 A.M.. (3) Max/Min operation (19%): the question demands finding the biggest or smallest number in a group. (4) False CoT (9%): the CoT generation model gives wrong or hallucinated reasoning steps, e.g., using numbers that do not exist in the table or the question when generating formulas. (5) Commonsense (7%): the commonsense knowledge is needed to answering the question, which is a weakness of small-scale language models.

## 5 Related Work

**CoT prompting for LLMs.** By providing a few in-context examples (or demonstrations) which contain chain-of-thoughts, CoT prompting can encourage LLMs to output intermediate reasoning steps before predicting the final answer (Wei et al., 2022). Existing CoT studies mainly focus on two directions. (1) Improving the quality of CoT demonstrations. For instance, selecting better in-context examples for CoT prompting according to the question diversity (Zhang et al., 2022), the solution complexity (Fu et al., 2023), or the example similarity (Rubin et al., 2022). (2) Exploring new representations of CoT reasoning steps. Beside the typical natural language format, researchers also proposed chain-of-thoughts in other formats. For instance, program-of-thoughts (Chen et al., 2022), tree-of-thoughts (Yao et al., 2023a), and graph-of-thoughts (Yao et al., 2023b). Among them, the CoT in program languages has emerged as a powerful approach for LLMs to invoking external tools (Qin et al., 2023). Recently, Lu et al. (2023a) proposed the Chameleon framework that augments LLMs with various tools like search engines and Python executors. We treat it as a contemporary work of our paper and list its results in the Appendix D.

**Pre-trained TaLMs.** Inspired by the success of pre-training on the natural language text, various TaLMs are proposed for pre-training on the semi-structured tabular data (Dong et al., 2022). Existing TaLMs mainly inherit the architectures of traditional language models and can be classified

into three types. (1) Encoder-based TaLMs like TAPAS (Herzig et al., 2020), MATE (Eisenschlos et al., 2021) and TUTA (Wang et al., 2021). (2) Encoder-Decoder TaLMs such as TAPEX (Liu et al., 2022) and STTP (Xing and Wan, 2021). (3) Decoder-based TaLMs like TableGPT (Gong et al., 2020). In previous studies, TaLMs are usually finetuned to directly generate final answers or simple formulas. By contrast, we are the first to explore the combination of the CoT reasoning and pre-trained TaLMs.

## 6 Conclusion

We extend the CoT reasoning into small-scale TaLMs for the first time, and provide an effective approach for tabular mathematical reasoning task, especially under scenarios where LLMs are not accessible. Specifically, we propose a novel framework named TaCo, which coordinates two TaLMs responsible for CoT generation and answer inference, respectively. By introducing an external calculator, we further augment TaCo with the accurate math computing ability. With two TAPEX-large models as backbones, the TaCo outperforms the state-of-the-art ChatGPT on the TABMWP dataset by 9.55% (82.60%→92.15%) with much less parameters (0.8B).

## Limitations

Though the proposed method achieves great performance with less parameters, the fine-tuning of the CoT generation model and the answer inference model depends on annotated chain-of-thoughts and gold answers. As a result, the chain-of-thought reasoning ability of TaCo could be limited to the tabular mathematical reasoning task. In the future research, one can utilize open-source LLMs to generate chain-of-thoughts of more diversities and of more table-related tasks (Wang et al., 2023b; Ho et al., 2023), which may further extend the generalization ability of TaLMs and reduce the cost of manual annotation.

In the aspect of external tools, compared with frameworks which enable LLMs to access various tools (Shen et al., 2023; Lu et al., 2023a), TaCo only utilizes a calculator to complete common arithmetic calculations, i.e., "+,-,×,÷". More advanced external tools may be integrated to enhance the capability of the framework. We believe that the tool learning with small-scale language models is a valuable future direction, especially for particular scenarios where LLMs are not available.

## Ethics Statement

This paper proposes a two-stage framework for the tabular mathematical reasoning task, and models are trained and evaluated on the public TABMWP dataset. Thus, the authors foresee no ethical concerns with the research in this paper.

## Acknowledgements

This work was supported by the National Natural Science Foundation of China (No. 61976207) and the National Social Science Foundation of China (No. 21AZD145).

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

## A  More Implementation Details

In our experiments, we employ TAPEX and Uni-fiedQA as backbones of TaCo framework. When linearizing the table into flattened sequence, if there exist no column headers in the original table, pseudo column headers will be inserted, e.g., 'Column header 1'. The hyper-parameter configurations of TAPEX and UnifiedQA backbone and their model sizes are shown in Table 6 and Table 7, respectively. Our experiments are all performed on a 32G NVIDIA V100 GPU.

For LLM-based baselines, we list numbers of few-shot examples and selection strategies in Table 8. For ChatGPT baseline, we randomly select 4 examples from train set for each question type. For fair comparison, we use the same prompt format as PromptPG (Lu et al., 2023b) to construct in-context examples, which is demonstrated in Figure 5.

| Parameters | TAPEX | |
|---|---|---|
| | base (140M) | large (400M) |
| Learning Rate | 3e-5 | 3e-5 |
| Batch Size | 16 | 32 |
| Weight Decay | 0.01 | 0.01 |
| Max Grad Norm | 1.0 | 1.0 |
| Warmup | Linear | Linear |
| Warmup Fraction | 0.1 | 0.1 |
| Epochs for Stage 1 | 20 | 25 |
| Epochs for Stage 2 | 15 | 20 |
| Training Time for Stage 1 | 3 hours | 8 hours |
| Training Time for Stage 2 | 2 hours | 6 hours |

Table 6: Hyper-parameter configurations for TAPEX backbone.

| Parameters | UnifiedQA | | |
|---|---|---|---|
| | small (60M) | base (220M) | large (770M) |
| Learning Rate | 5e-5 | 5e-5 | 5e-5 |
| Batch Size | 16 | 16 | 48 |
| Weight Decay | 0.01 | 0.01 | 0.01 |
| Max Grad Norm | 1.0 | 1.0 | 1.0 |
| Warmup | Linear | Linear | Linear |
| Warmup Fraction | 0.1 | 0.1 | 0.1 |
| Epochs for Stage 1 | 15 | 20 | 25 |
| Epochs for Stage 2 | 15 | 15 | 20 |
| Training Time for Stage 1 | 2 hours | 8 hours | 15 hours |
| Training Time for Stage 2 | 2 hours | 5 hours | 12 hours |

Table 7: Hyper-parameter configurations for UnifiedQA backbone.

| Method | # few-shot examples | Selection strategy | Acc-Test |
|---|---|---|---|
| GPT-3 | 2 | Random selection | 57.13 |
| Codex | 4 | Manual construction | 59.40 |
| GPT-3+CoT | 2 | Random selection | 62.92 |
| Codex+CoT | 4 | Manual construction | 65.20 |
| PromptPG | 2 | Policy Gradient | 68.23 |
| PoT | 4 | Manual construction | 73.20 |
| ChatGPT | 4 | Random selection | 65.52 |
| ChatGPT+CoT | 4 | Random selection | 82.60 |

Table 8: Number of in-context examples and selection strategies of LLM baselines.

## B  The complexity of CoT generation

Table 3 reveals a significant performance difference between *free-text* questions and *multi-choice* questions. To shed more light on the TABMWP dataset, we quantitatively analyze the complexity of the CoT generation for two question types. Specifically, we compute the number of required numerical calculations in the gold CoT (including +, -, ×, ÷, counting, min, max), the number of reasoning steps (we treat each line in the gold CoT as one reasoning step for simplicity) and the length of the gold CoT. The statistical results in the Table 9 demonstrate that, in the TABMWP dataset, the CoT generation from *free-text* questions is more complex than that from *multi-choice* questions. Based on our observations, at least 18% *multi-choice* questions (mainly of EXTR and OTH answer types) do not need numerical calculations, but almost all *free-text* questions need numerical calculations.

## C  Error Instances and More Analysis

In this section, we present detailed error instances to analyze the weakness of TaCo framework, which is shown in Figure 7 to Figure 10. We find that most of errors are caused by the inability of used external tool and the representation of chain-of-thoughts. Take the error instance in Figure 7 as an example. To correctly answer the question in Figure 7, the model should find numbers from the table which are greater than 53, and then count how many numbers are found. However, as the CoT generation model is fine-tuned to generate chain-of-thoughts in simple natural language, it is difficult for the model to describe the above process in a short and straightforward expression, which makes it hard to invoke external tools. If we could represent chain-of-thoughts in program languages like Python, the solution of this question would be much more clear. For instance, one can write a line of Python code: "`Ans = Count(61,61,65,65,66,70,66,78)`", and implement a Python function "`Count()`" as an external tool to get the accurate result. The same methodology could be applied to error instances which demand other abilities such as fraction calculation, min/max operation and time calculation. Besides, lacking commonsense knowledge also increases the difficulty for models to comprehend tables and questions, e.g., reading bus schedule in Figure 10.

| | |
|---|---|
| **Table:** | |
| apron \| $20.00 | |
| soup pot \| $45.00 | |
| rolling pin \| $12.00 | |
| set of forks \| $13.00 | |
| serving platter \| $10.00 | |
| cookbook \| $16.00 | |

| apron | $20.00 |
|---|---|
| soup pot | $45.00 |
| rolling pin | $12.00 |
| set of forks | $13.00 |
| serving platter | $10.00 |
| cookbook | $16.00 |

**Question:** How much money does Noah need to buy an apron and a soup pot? (Unit: $)

**Answer:**

Add the price of an apron and the price of a soup pot:

$20.00 + $45.00 = $65.00

Noah needs $65.00.

The answer is 65.

Figure 5: The format of in-context examples for ChatGPT baseline (ID:19324).

| Question Types | # of numerical calculations (median/mean) | # of reasoning steps (median/mean) | the length of CoT (median/mean) |
|---|---|---|---|
| *free-text* | 2.00/2.15 | 4.00/5.18 | 196.00/239.15 |
| *multi-choice* | 1.00/1.78 | 2.00/3.84 | 180.00/253.21 |

Table 9: The quantitative analysis of the complexity of the CoT generation for two question types.

## D Results of Chameleon framework

Recently, Lu et al. (2023a) proposed a compositional reasoning framework named Chameleon, which treats LLMs as a natural language planner to utilize a variety of tools including vision models, web search engines, Python functions and so on. As shown in Table 10, based on the powerful GPT-4 and multiple external tools, Chameleon achieves the best accuracy of 98.78% on TABMWP test set. However, the proposed TaCo framework still achieves a competitive result of 92.15% with less parameters.

We also apply the same calculator to the output of ChatGPT and use regular expressions to extract the final answer from the output. There is a slight performance increase from 82.60% to 83.07%. After inspecting error cases of ChatGPT, we found that most errors resulted from wrong reasoning steps rather than calculation mistakes. Compared with small-scale TaLMs, the numerical calculating ability of ChatGPT is much more better, which may attribute to the potential use of more advanced external tools behind the ChatGPT system.

| Method | Acc-Test | Question Types | |
|---|---|---|---|
| | | FREE | MC |
| ChatGPT CoT | 82.03 | 78.43 | 92.32 |
| ChatGPT PoT | 89.49 | 90.24 | 87.35 |
| GPT-4 CoT | 90.81 | 88.48 | 97.49 |
| GPT-4 PoT | 96.93 | 97.40 | 95.58 |
| Chameleon (ChatGPT) | 93.28 | 93.13 | 93.72 |
| Chameleon (GPT-4) | 98.78 | 98.95 | 98.29 |
| TaCo (Ours) | 92.15±0.13 | 91.69 | 93.47 |

Table 10: Accuracy of Chameleon on TABMWP test set.

| | |
|---|---|
| caramel corn | $1.13 per kg |
| chocolate popcorn | $1.78 per kg |
| kettle corn | $0.69 per kg |
| ranch popcorn | $1.72 per kg |
| cheddar popcorn | $0.90 per kg |
| Parmesan popcorn | $1.41 per kg |

**Question:** How much would it cost to buy 3 kilograms of kettle corn and 2 kilograms of Parmesan popcorn? (Unit: $)

**Generated Solution:** Find the cost of the kettle corn. Multiply:
$0.69 × 3 = **$2.07**.
Find the cost of the Parmesan popcorn. Multiply:
$1.41 × 2 = **$2.82**.
Now find the total cost by adding:
**$2.07** + **$2.82** = **$4.89**. It would cost **$4.89**.
**Predicted Answer**: **4.89**

Figure 6: A correct instance where TaCo generates right solution and answer. (ID:752).

**Broken tiles per shipment**

| Stem | Leaf |
|---|---|
| 2 | 9 |
| 3 | 1 6 8 9 |
| 4 | 1 2 3 7 9 9 |
| 5 | 0 |
| 6 | 1 1 3 5 6 |
| 7 | 0 6 8 |

**Question:** Clarksville Flooring tracked the number of broken tiles in each shipment it received last year. How many shipments had at least 53 broken tiles? (Unit: shipments)

**Solution of TaCo:**
Find the row with stem 5.
Count all the leaves greater than or equal to 3.
Count all the leaves in the rows with stems 6 and 7.
You counted **9** leaves.
**9** shipments had at least 53 broken tiles.
**Answer of TaCo**: **9** shipments  **Correct Answer**: **8**

Figure 7: An error instance of counting operation (ID:449), where TaCo cannot correctly count how many numbers satisfying requirements.

**Drink preferences**

| Drink | Number of students |
|---|---|
| Diet cola | 450 |
| Fruit punch | 680 |
| Lemon-lime soda | 130 |
| Other | 320 |

**Question:** Students at Fairfax High School were polled regarding their favorite drinks. What fraction of students preferred fruit punch? Simplify your answer.

**Solution of TaCo:**
Find how many students preferred fruit punch: 680
Find how many students were polled in total: 450 + 680 + 130 + 320 = 1580
Divide 680 by 1580: $\frac{680}{1580}$
Reduce the fraction: $\frac{44}{58}$
$\frac{44}{58}$ of students preferred fruit punch..
**Answer of TaCo**: $\frac{44}{58}$  **Correct Answer**: $\frac{35}{79}$

Figure 8: An error instance of fraction calculation (ID:1711), where TaCo makes mistakes when reducing a fraction.

| | |
|---|---|
| play ticket | $24.00 |
| comedy show ticket | $49.00 |
| boat show ticket | $23.00 |
| soccer ticket | $50.00 |

**Question:** Dominic has $72.00. Does he have enough to buy a play ticket and a soccer ticket?
**Option:** (A) yes (B) no

**Solution of TaCo:**
Add the price of a play ticket and the price of a soccer ticket:
$24.00 + $50.00 = $74.00
Since Dominic has $72.00, he has just enough money.
**Answer of TaCo**: **yes**    **Correct Answer**: **No**

Figure 9: An error instance of number comparison (ID:1434), where TaCo cannot correctly judge which is the larger number between 72.00 and 74.00.

**Bus schedule**

| | | | | | |
|---|---|---|---|---|---|
| the kickball field | 8:45 A.M. | 9:15 A.M. | 10:15 A.M. | 10:30 A.M. | 10:45 A.M. |
| the library | 9:00 A.M. | 9:30 A.M. | 10:30 A.M. | 10:45 A.M. | 11:00 A.M. |
| the school | 9:30 A.M. | 10:00 A.M. | 11:00 A.M. | 11:15 A.M. | 11:30 A.M. |
| the mall | 10:30 A.M. | 11:00 A.M. | 12:00 P.M. | 12:15 P.M. | 12:30 P.M. |
| the zoo | 11:30 A.M. | 12:00 P.M. | 1:00 P.M. | 1:15 P.M. | 1:30 P.M. |
| the playground | 12:00 P.M. | 12:30 P.M. | 1:30 P.M. | 1:45 P.M. | 2:00 P.M. |
| the doctor's office | 12:30 P.M. | 1:00 P.M. | 2:00 P.M. | 2:15 P.M. | 2:30 P.M. |
| the post office | 12:45 P.M. | 1:15 P.M. | 2:15 P.M. | 2:30 P.M. | 2:45 P.M. |
| the science museum | 1:00 P.M. | 1:30 P.M. | 2:30 P.M. | 2:45 P.M. | 3:00 P.M. |

**Question:** Look at the following schedule. Ruth just missed the 11.00 A.M. bus at the mall. How long does she have to wait until the next bus?
**Option:** (A) 15 minutes (B) 30 minutes (C) 45 minutes (D) 1 hour

**Solution of TaCo:**
Find 11:00 A. M. in the row for the mall.
Look for the next bus in that row.
The next bus is at 12:00 P. M.
Find the elapsed time. The elapsed time is **30** minutes.
**Answer of TaCo**: **30 minutes.**    **Correct Answer**: **1 hour**

Figure 10: An error instance of time calculation (ID:2766), where TaCo fails to compute the elapsed time between 11:00 A.M. and 12:00 P.M.