# OpenReview forum: "Chain-of-Thought Reasoning in Tabular Language Models"
_EMNLP/2023/Conference — EMNLP 2023 Findings_

### Official Review · Reviewer_Z9oN · 2023-08-05

**Soundness:** 3

**Excitement:**

3: Ambivalent: It has merits (e.g., it reports state-of-the-art results, the idea is nice), but there are key weaknesses (e.g., it describes incremental work), and it can significantly benefit from another round of revision. However, I won't object to accepting it if my co-reviewers champion it.

**Paper Topic And Main Contributions:**

This paper proposes TaCo, which utilizes two tabular LMs, one for CoT generation, and another one for answer inference.  This can be regarded as a combination of the method of CoT with traditional tabular-related LMs.

Main contributions include:
1. the first to explore cot reasoning in tabular-related tasks.
2. achieve significant performance improvement.

**Questions For The Authors:**

1. Could you explain the value of the answer inference model? Can we discard this model? Is this setting the same as QT -> SA in the paper?
2. What if the training dataset does not contain detailed solutions? A more general method is highly expected since most datasets do not have detailed solutions.
3. Could you add more benchmark datasets to cross-validate your claims?

**Reasons To Accept:**

1. A good initiative to start the research: a. pre-trained small tabular LMs perform worse in cot reasoning ability. b. tabular LMs benefit from cot when inferencing the final answer.
2. thorough experiments showing good results.


**Reasons To Reject:**

1. incremental work. Look like the same method as previous cot work applied to the table domain.
2. expect a more general method to adapt to those datasets that don't have detailed solutions (cot).

**Reproducibility:**

4: Could mostly reproduce the results, but there may be some variation because of sample variance or minor variations in their interpretation of the protocol or method.

**Reviewer Confidence:**

4: Quite sure. I tried to check the important points carefully. It's unlikely, though conceivable, that I missed something that should affect my ratings.

---

> ### Author Rebuttal · Authors · 2023-08-29
>
> We sincerely thank you for the careful review and insightful feedback. Your constructive suggestions are crucial for us to refine our work. We are also delighted to learn that you regarded our work as "A good initiative to start the research" and considered experiments "thorough and showing good results". We hope our response could alleviate your concerns and your valuable advice will be fully incorporated in the final version.
>
> **Q1: Incremental work. Look like the same method as previous cot work applied to the table domain.**
>
> **A1:** Thank you for this constructive comment. Admittedly, LLMs and CoT reasoning have experienced rapid advancements since the last year. Different CoT prompting methods have been explored to elicit intermediate reasoning steps from LLMs to solve complex questions[1-5]. Nevertheless, most of previous CoT work is based on LLMs and mainly focuses on unstructured text data. This raises two natural questions: **(i) Does this mean the end of traditional small-scale LMs? (ii) On the structured and prevalent table data, how good are the CoT reasoning abilities of existing LMs?**
>
> **The contribution of our work lies exactly on the intersection of above two research scopes which have not been thoroughly investigated.** To summarize, the contribution of our work are threefold. (1) We are the first to extend the CoT reasoning ability into TaLMs and advocate a general two-stage paradigm, which could be combined with future TaLMs. (2) The proposed method TaCo outperforms powerful LLM baselines with much less parameters and can surve as an alternative approach for scenarios where LLM-based methods are not suitable. (3) Thorough experiments were conducted to present a comprehensive evaluation and meaningful insights. As a result, we respectfully argue that the contribution of our work is not incremental.
>
> Moreover, we believe that the CoT reasoning on the tabular data is a promising direction and it deserves further exploring. For example, designing in-context learning strategy which considers table structures, enhancing the TaLM's CoT reasoning ability across different table types. We hope our work could attract more interests into this field.
>
> **Q2: The value of the answer inference model.**
>
> **A2:** Thank you for this insightful question. In the following analysis, we demonstate two important effect of Answer Inference model with detailed case studies.
>
> 1) **Answer Inference model handles different CoTs with flexible answer locations and various answer types.** If we directly use regular expressions to extract the final answer from the generated CoT (we can denote this method as "QT-->S"), it is very difficult to cover different cases where multi-type answers may occur at arbitrary locations, which are demonstrated in the following CoTs with bold answer text.
>
> > … Estelle will get to Princeton at **10:30 P. M.** (ID: 1105)
> >
> > … the constant of proportionality is **2** pounds per plate. (ID: 1136)
> >
> > … **2** boxes had at least 44 broken crayons but fewer than 54 broken crayons. (ID: 1326)
> >
> > ... In 1995, **cantaloupes** cost more. (ID: 2394)
> >
> > ... An airplane calendar costs $**3.13** more than a map of Africa. (ID: 2643)
>
> Thus, it is desirable to train an Answer Inference model which can automatically infer final answers based on different CoTs.
>
> 2) **Answer Inference model brings a second chance to recontemplate improved reasoning steps before making the final judgement.** For the "QT-->SA" method, we construct a concluding sentence "As a result, the final answer is {answer_text}" and append it to the end of CoT, which is used to train the TaLM to generate reasoning steps and the final answer in one forward pass.  Then, the regular expression is adopted to extract the final answer from the model output:
>
> ```python
> import re
> ret = re.match('.*the final answer is (.*)', model_output)
> final_answer = ret.group(1)
> ```
>
> The "QT-->SA" indeed reduces the difficulty to extract the final answer. However, **such one-stage paradigms cannot fully utilize the corrected CoT to change the original (wrong) answer.** In the following example (ID: 22198), though errors in the CoT are corrected by the external calculator, the original wrong answer is not changed. By contrast,  **the two-stage paradigm "QT-->S-->A" can take full advantage of the corrected CoT to infer the right answer instead of jumping to the conclusion directly.**
>
> >**Question:** Shelley has $272.00. Does she have enough to buy a lawn mower and an iron bench? (A) yes (B) no
> >
> >**Original Output of QT->SA:**
> >
> > Add the price of a lawn mower and the price of an iron bench:
> >
> > 145.00 + 140.00 = \$265.00
> >
> > As a result, the final answer is yes.
> >
> >**Corrected Output of QT->SA:**
> >
> > Add the price of a lawn mower and the price of an iron bench:
> >
> > 145.00 + 140.00 = \$285.00
> >
> > As a result, the final answer is yes.
> >
> >**Final Answer of QT->SA:** yes
> >
> >**Correct Answer:** no
>
> In addition, **it is challenging for one small-scale TaLM to conduct both CoT reasoning and answer inferring**. For example (ID: 1937), the "QT->SA" chooses the wrong number as the final answer more often than "QT->S->A".
>
> > **Output of QT->SA:** ... 5 amusement parks have at least 10 roller coasters but fewer than 40 roller coasters.
> As a result, the final answer is 10.
> >
> >**Final Answer of QT->SA:** 10
> >
> >**Correct Answer:** 5
>
> **The similar two-stage paradigm has also been adopted in recent work** like Verify-and-Edit[1] and Self-ask[2], where they first utilize one LLM to generate the CoT and then ask the same LLM to infer the final answer based on the improved CoT. By contrast, we delegate CoT generation and answer inference to two small TaLMs, respectively. In the next version of the paper, more case studies will be included to highlight the importance of Answer Inference model.
>
>
> **Q3: A more general method is highly expected since most datasets do not have detailed solutions.**
>
> **A3:** We fully understand that a more general method which needs the fewest human-annotated CoTs is highly expected. **As mentioned in the Limitation Section, for datasets lacking annotated solutions, we can utilize LLMs to generate CoTs[8-9] and then use them to further enhance the generalization of TaCo**. This can be viewed as a kind of transfer learning from LLMs to TaLMs and it can greatly alleviate the dependency on manual annotations.
>
> **Q4: Adding more benchmark datasets to evaluate the proposed method.**
>
> **A4:** Thank you for this great advice. We select the TABMWP dataset to conduct experiments as it is the latest large-scale table dataset with high-quality gold solutions. More experiments on other datasets are promised. We will utilize the transfer learning method in A3 to train the TaCo on more datasets and evaluate its performance.
>
>
> [1] Chain-of-Thought Prompting Elicits Reasoning in Large Language Models. NIPS 2022.
>
> [2] Large Language Models are Zero-Shot Reasoners. NIPS 2022.
>
> [3] Self-Consistency Improves Chain of Thought Reasoning in Language Models. ICLR 2023.
>
> [4] Least-to-Most Prompting Enables Complex Reasoning in Large Language Models. ICLR 2023.
>
> [5] Complexity-Based Prompting for Multi-Step Reasoning. ICLR 2023.
>
> [6] Verify-and-Edit: A Knowledge-Enhanced Chain-of-Thought Framework. ACL 2023.
>
> [7] Measuring and Narrowing the Compositionality Gap in Language Models. 2022.
>
> [8] Large Language Models Are Reasoning Teachers. ACL 2023.
>
> [9] Self-Instruct: Aligning Language Models with Self-Generated Instructions. ACL 2023.

---

### Official Review · Reviewer_xSdW · 2023-08-05

**Soundness:** 4

**Excitement:**

4: Strong: This paper deepens the understanding of some phenomenon or lowers the barriers to an existing research direction.

**Paper Topic And Main Contributions:**

The authors present a new two-stage model for tabular QA where each stage is implemented with a small-scale tabular language model. The first stage produces a CoT and the second stage outputs the answer. The demonstrate that their proposed method achieves state-of-the-art performance on the TABMWP dataset.

**Questions For The Authors:**

 A. In the zero-shot pilot experiments, was TAPEX fine-tuned any further on zero-shot examples, or did you use the existing fine-tuned model?

 B. TaCo outperforms baselines moreso on questions with integer or decimal answers. How much of this performance gap is accounted by the external calculator? Is it feasible to apply the external calculator correction to the outputs of the LLMs? If you've already tried this, is there a notable difference in accuracy?

 C. What is the "heuristic guess" method listed in Table 3?

**Reasons To Accept:**

 - The proposed model is interesting and performs well empirically, and is much more parameter-efficient than the baseline LMs.
 - There are a number of well-thought ablation experiments that help to to highlight which components of the model are most instrumental.

**Reasons To Reject:**

 - Reproducibility would be improved if the experiments included non-GPT open LMs. But since the LMs were only used as baselines, this is not a huge concern.
 - The presentation could be made cleaner.

**Reproducibility:**

5: Could easily reproduce the results.

**Reviewer Confidence:**

4: Quite sure. I tried to check the important points carefully. It's unlikely, though conceivable, that I missed something that should affect my ratings.

**Typos Grammar Style And Presentation Improvements:**

There are a small number of minor errors. I list some of the error I found in the first few pages, but I encourage the authors to read through the entire manuscript more carefully and correct all such errors.

Lines 57-58: Redundant use of "requirements."

Line 65: "despite" -> "despite the fact that"

Line 93: "Experiment" -> "Experimental"

Line 102: "We" -> "we"

Line 151: "golden CoT" -> "Gold CoT" This sentence can also be rephrased to be clearer.

Figure 2c: The verb "exploiting" is usually used with a direct object.

Line 195: "flatten" -> "flattened"

Line 207: "resulted" -> "resulting"

There are also a number of examples of noun phrases with missing determiners (e.g. "the", "a", etc).

Also, it seems that the term "solution" is used later in the paper to refer to CoTs. This may be confusing for readers since colloquially, "solution" is often synonymous with "answer."

---

> ### Author Rebuttal · Authors · 2023-08-29
>
> We sincerely thank you for the valuable effort and the precious time spent for carefully reviewing our paper. We greatly appreciate your positive feedback and we are very glad to learn that you found our work "interesting" and "well-thought". Your thoughtful and valuable suggestions will be incorporated in the future version of our paper. The questions and problems will be properly addressed as we reply.
>
> **Q1: Including more non-GPT open LMs as baselines.**
>
> **A1:** Thank you for this constructive advice. The non-GPT open LMs like LLaMA[1] and Alpaca[2] have played an important role in the recent open-source AI community. We will consider evaluating their CoT reasoning ability on the tabular data in the future work.
>
> **Q2: In the zero-shot pilot experiments, was TAPEX fine-tuned any further on zero-shot examples, or did you use the existing fine-tuned model?**
>
> **A2:** We follow the TABMWP paper and use the fine-tuned TAPEX as a strong backbone to conduct pilot experiments. **It was pre-trained and then fine-tuned on the WTQ[3] dataset, with no further fine-tuning on zero-shot examples.** Model checkpoints were directly downloaded from the offical huggingface model hub ("tapex-base-finetuned-wtq" and "tapex-large-finetuned-wtq"). More details will be included in the future version.
>
> **Q3:  How much of the performance gap on questions with integer or decimal answers is accounted by the external calculator?**
>
> **A3:** The performance of TaCo without the external calculator is listed below, where accuracy scores on test questions with integer (Acc-INT) or decimal (Acc-DEC) answers are reported respectively. **The experimental results indicate that the external calculator is an essential contributor for INT and DEC questions, which accouts for an average performance gap of about 35%.** Compared with INT questions, the performance gap on DEC questions is more significant as they usually demand more intensive and accurate numerical calculations.
>
> | **Model**          | **Acc-INT** | **Acc-DEC** | **Ave. Acc Gap** |
> | :----------------: | :---------: | :---------: | :---------------: |
> | TaCo  (base)       | 85\.29      | 86\.44      | -                 |
> | TaCo  (large)      | 92\.54      | 88\.41      | -                 |
> | **w/o calculator** |             |             |                   |
> | QT → S → A (base)  | 62\.95      | 30\.13      | 39\.325           |
> | QT → S → A (large) | 73\.70      | 44\.72      | 31\.265           |
>
> **Q4: Is it feasible to apply the external calculator correction to the outputs of the LLMs?**
>
> **A4:** Yes. We applied the same calculator to the output of ChatGPT and use regular expressions to extract the final answer from the output. **There is a slight performance increase from 82.60% to 83.07%.** After inspecting error cases of ChatGPT, we found that most errors resulted from wrong reasoning steps rather than calculation mistakes. Compared with small-scale TaLMs, the numerical calculating ability of ChatGPT is much more better. Thus the effect of the external calculator is not significant.
>
> However, we believe that the "Tool Learning with LLMs"[4] is a very promising direction. Various advanced tools like programming languages and search engines have been combined with LLMs and have shown great performance on different tasks including the tabular mathematical reasoning[5-6].
>
> **Q5: What is the "heuristic guess" method listed in Table 3?**
>
> **A5:** We are sorry that related descriptions were missing and they will be added in the final version. The "heuristic guess" is a baseline from the TABMWP paper. For _multi-choice_ questions, it randomly selects one from the given options with even probabilities. For _free-text_ questions, it randomly chooses one number from the question or the table as the prediction.
>
> **Q6: Typos, Grammar, Style And Presentation Improvements**
>
> **A6:** Once again, we would like to express our sincere gratitude for your careful review and precious suggestions, which remarkably improve the quality of our paper. We will incorpate all these changes in the final revision and read through the entire manuscript to refine the presentation.
>
> [1] LLaMA: Open and Efficient Foundation Language Models. 2023.
>
> [2] Alpaca: A Strong, Replicable Instruction-Following Model. 2023.
>
> [3] Compositional Semantic Parsing on Semi-Structured Tables. ACL 2015.
>
> [4] Tool Learning with Foundation Models. 2023.
>
> [5] Program of Thoughts Prompting: Disentangling Computation from Reasoning for Numerical Reasoning Tasks. 2022.
>
> [6] Chameleon: Plug-and-Play Compositional Reasoning with Large Language Models. 2023.

---

### Official Review · Reviewer_UoLv · 2023-08-05

**Soundness:** 3

**Excitement:**

3: Ambivalent: It has merits (e.g., it reports state-of-the-art results, the idea is nice), but there are key weaknesses (e.g., it describes incremental work), and it can significantly benefit from another round of revision. However, I won't object to accepting it if my co-reviewers champion it.

**Paper Topic And Main Contributions:**

Topic
Corporate chain-of-thought reasoning in small-scale tabular language models for real use case (with resource limitation and data security)

Main contribution
1. Propose a new paradigm of tabular mathematical reasoning
2. Propose a framework TaCo which uses 2 small-scale TaLMs for CoT generation and answer inference respectively and also has mechanism to deal with incorrect calculation by using string parsing and external calculator.
3. Achieve SoTA result in the evaluation benchmark, along with an extensive analysis of result.


**Questions For The Authors:**

1. In (3) of subsection 4.3, I am not so sure why only free-text questions requires the model to perform complicated numerical calculation. As to answer the question, TaCo always generate the CoT first, then distill into the final answer. Thus, from my intuition, the complexity of the CoT generation in two setting free-text or CoT is roughly similar. Please elborate it.

2. About the effect of two-stage paradigm, from my point of view, the role of Answer Inference Model is not significant. In fact, a piece of hard code using regex can cover almost all cases of output to extract the final answer. Thus, I am skeptical about the result of QT -> SA in Table 4. To make the claim about the importance of Answer Inference Model more convincing, some case studies are needed (better if the code/algo to extract final answer is provided).

**Reasons To Accept:**

1. The organization of this paper is quite interesting. It shows the pilot the study as the motivation to conduct a larger scale study on the tabular mathematical reasoning problem, which better equips readers with the context of this research.
2. It conducts a very extensive evaluation, comparing its models with a lot of other settings. The paper also analyzes the result and ablates the model in different aspects, giving readers many insights about the model.

**Reasons To Reject:**

Please refer to questions.

**Reproducibility:**

3: Could reproduce the results with some difficulty. The settings of parameters are underspecified or subjectively determined; the training/evaluation data are not widely available.

**Reviewer Confidence:**

3: Pretty sure, but there's a chance I missed something. Although I have a good feel for this area in general, I did not carefully check the paper's details, e.g., the math, experimental design, or novelty.

---

> ### Author Rebuttal · Authors · 2023-08-29
>
> Thank you very much for the thorough review and valuable comments. We are delighted to learn that our paper "gives readers many insights" together with "an interesting organization" which better equips readers with the research context. Moreover, we are pleased to get your constructive feedback which is very useful to further improve our work. We hope the following responses will adequately address your concerns.
>
> **Q1: Why only _free-text_ questions require the model to perform complicated numerical calculations?**
>
> **A1:** We are sorry that some expressions in Section 4.3 (3) result in some confusion and ambiguity. In fact, we intended to point out that, **compared with _multi-choice_ questions, _free-text_ questions in the TABMWP dataset usually require more complicated numerical calculations,** which makes it more difficult to obtain correct answers.
>
> The evidence is as follows.
>
> 1) _Free-text_ questions include two answer types, i.e., integer answers (INT) and decimal answers (DEC), and _multi-choice_ questions include three answer types, i.e., extractive text answers (EXTR), boolean text answers (BOOL) and other text answers (OTH). Based on our observations, at least 18% _multi-choice_ questions (mainly of EXTR and OTH answer types) do not need numerical calculations, but almost all _free-text_ questions need numerical calculations.
>
> 2) **We quantitatively analyze the complexity of the CoT generation for two question types.** We compute the number of required numerical calculations in CoT (including +, -, ×, ÷, counting, min, max), the number of reasoning steps in CoT (we treat one line as one reasoning step for simplicity) and the length of CoT. The results are shown below.
>
> | Question Types | # of numerical calculations  (median/mean) | # of reasoning steps (median/mean) | the length of CoT  (median/mean)  |
> | :---: | :---: | :---: | :---: |
> | _free-text_ | 2.00/2.15 | 4.00/5.18 | 196.00/239.15 |
> | _multi-choice_ | 1.00/1.78 | 2.00/3.84 | 180.00/253.21 |
>
> Statistical results reveal that, in the TABMWP dataset, the CoT generation from _free-text_ questions is more complex than that from _multi-choice_ questions, which finally leads to different model performance. Related expressions in the paper will be revised to make them more clear and above evidence will be added to support our claim.
>
> **Q2: The role of Answer Inference Model is not significant. It can be replaced by regular expressions.**
>
> **A2:** Thank you for this valuable question. In the following analysis, we showcase two important effect of Answer Inference model.
>
> 1) **Answer Inference model handles different CoTs with flexible answer locations and various answer types.** If we directly use regular expressions to extract the final answer from the generated CoT (we can denote this method as "QT-->S"), it is very difficult to cover different cases where multi-type answers may occur at arbitrary locations, which are demonstrated in the following CoTs with bold answer text.
>
> > … Estelle will get to Princeton at **10:30 P. M.** (ID: 1105)
> >
> > … the constant of proportionality is **2** pounds per plate. (ID: 1136)
> >
> > … **2** boxes had at least 44 broken crayons but fewer than 54 broken crayons. (ID: 1326)
> >
> > ... In 1995, **cantaloupes** cost more. (ID: 2394)
> >
> > ... An airplane calendar costs $**3.13** more than a map of Africa. (ID: 2643)
>
> Thus, it is desirable to train an Answer Inference model which can automatically infer final answers based on different CoTs.
>
> 2) **Answer Inference model brings a second chance to recontemplate improved reasoning steps before making the final judgement.** For the "QT-->SA" method, we construct a concluding sentence "As a result, the final answer is {answer_text}" and append it to the end of CoT, which is used to train the TaLM to generate reasoning steps and the final answer in one forward pass.  Then, the regular expression is adopted to extract the final answer from the model output:
>
> ```python
> import re
> ret = re.match('.*the final answer is (.*)', model_output)
> final_answer = ret.group(1)
> ```
>
> The "QT-->SA" indeed reduces the difficulty to extract the final answer. However, **such one-stage paradigms cannot fully utilize the corrected CoT to change the original (wrong) answer.** In the following example (ID: 22198), though errors in the CoT are corrected by the external calculator, the original wrong answer is not changed. By contrast,  **the two-stage paradigm "QT-->S-->A" can take full advantage of the corrected CoT to infer the right answer instead of jumping to the conclusion directly.**
>
> >**Question:** Shelley has $272.00. Does she have enough to buy a lawn mower and an iron bench? (A) yes (B) no
> >
> >**Original Output of QT->SA:**
> >
> > Add the price of a lawn mower and the price of an iron bench:
> >
> > 145.00 + 140.00 = \$265.00
> >
> > As a result, the final answer is yes.
> >
> >**Corrected Output of QT->SA:**
> >
> > Add the price of a lawn mower and the price of an iron bench:
> >
> > 145.00 + 140.00 = \$285.00
> >
> > As a result, the final answer is yes.
> >
> >**Final Answer of QT->SA:** yes
> >
> >**Correct Answer:** no
>
> In addition, **it is challenging for one small-scale TaLM to conduct both CoT reasoning and answer inferring**. For example (ID: 1937), the "QT->SA" chooses the wrong number as the final answer more often than "QT->S->A".
>
> > **Output of QT->SA:** ... 5 amusement parks have at least 10 roller coasters but fewer than 40 roller coasters.
> As a result, the final answer is 10.
> >
> >**Final Answer of QT->SA:** 10
> >
> >**Correct Answer:** 5
>
> **The similar two-stage paradigm has also been adopted in recent work** like Verify-and-Edit[1] and Self-ask[2], where they first utilize one LLM to generate the CoT and then ask the same LLM to infer the final answer based on the improved CoT. By contrast, we delegate CoT generation and answer inference to two small TaLMs, respectively. In the next version of the paper, more case studies will be included to highlight the importance of Answer Inference model.
>
> **Q3: Reproducibility problem, e.g., The settings of parameters are underspecified or subjectively determined.**
>
> **A3:** Due to space limitation, implementation details including parameter settings are listed in the Section 4.2 and Appendix A. We will add more implementation details about ablation studies and the code will be released along with the paper.
>
> [1] Verify-and-Edit: A Knowledge-Enhanced Chain-of-Thought Framework. ACL 2023.
>
> [2] Measuring and Narrowing the Compositionality Gap in Language Models. 2022.

---

### Meta-Review · Area_Chair_xvRR · 2023-09-19

**Recommendation:** 4

**Metareview:**

This work proposes a two stage method for tabular reasoning and calculations. The two-stage involves using smaller language models to reduce the cost of running larger language models for this task. The reviewers highlighted the comprehensive list of experiments and ablations and novelty of exploring CoT for the task. Reviewers have suggested some revision to improve the presentation of the work which should be considered by the authors and community would benefit from more clear presentation. Also since one of the main contribution of the work is the computational cost it would be good to discuss this along the dimensions of accuracy. It is good to see the limitation section in the manuscript, where authors discussed the limitations including reliance on CoT training data which might not be available in more general settings.

---

### Decision · Program_Chairs · 2023-10-07

**Decision:**

Accept-Findings

**Comment:**

This work proposes a two stage method for tabular reasoning and calculations. The two-stage involves using smaller language models to reduce the cost of running larger language models for this task. The reviewers highlighted the comprehensive list of experiments and ablations and novelty of exploring CoT for the task. Reviewers have suggested some revision to improve the presentation of the work which should be considered by the authors and community would benefit from more clear presentation. Also since one of the main contribution of the work is the computational cost it would be good to discuss this along the dimensions of accuracy. It is good to see the limitation section in the manuscript, where authors discussed the limitations including reliance on CoT training data which might not be available in more general settings.